# Scaling Safe Multi-Agent Control for Signal Temporal Logic Specifications

**Joe Eappen, Zikang Xiong, Dipam Patel, Aniket Bera, Suresh Jagannathan**
Purdue University
West Lafayette, IN 47907, United States
{jeappen,xiong84,dipam,aniketbera,sjaganna}@purdue.edu

**Abstract:** Existing methods for safe multi-agent control using logic specifications like Signal Temporal Logic (STL) often face scalability issues. This is because they rely either on single-agent perspectives or on Mixed Integer Linear Programming (MILP)-based planners, which are complex to optimize. These methods have proven to be computationally expensive and inefficient when dealing with a large number of agents. To address these limitations, we present a new scalable approach to multi-agent control in this setting. Our method treats the relationships between agents using a graph structure rather than in terms of a single-agent perspective. Moreover, it combines a multi-agent collision avoidance controller with a Graph Neural Network (GNN) based planner, models the system in a decentralized fashion, and trains on STL-based objectives to generate safe and efficient plans for multiple agents, thereby optimizing the satisfaction of complex temporal specifications while also facilitating multi-agent collision avoidance. Our experiments show that our approach significantly outperforms existing methods that use a state-of-the-art MILP-based planner in terms of scalability and performance.

**Keywords:** Multi-Robot Systems, Path Planning for Multiple Mobile Robots, Collision Avoidance, Specification-Guided Learning, Deep Learning Methods

## 1 Introduction

Learning-based methods have shown promise in multi-agent systems (MAS) for tasks such as collision avoidance, path planning, and task allocation [1, 2, 3, 4]. Extensions have also been developed to handle complex temporal tasks that may be described using formal languages such as Signal Temporal Logic (STL) [5, 6] and other temporal logics [7, 8, 9]; unfortunately, these methods have well-known limitations in terms of scalability and performance.

Signal Temporal Logic (STL) is a formal language for specifying complex temporal tasks that can be used to describe the behavior of agents in a multi-agent system. In many settings, including autonomous vehicles [10], drones [11], and robotic swarms [12], it is essential to ensure that the agents satisfy complex temporal tasks such as sequentially visiting a series of locations while avoiding collisions with each other and the environment. Once the user has specified the task in STL, the task can be synthesized using formal methods [13, 14, 15] in certain environments; however, these methods often struggle to scale to complex specifications and environments. In response to these challenges, Mixed Integer Linear Programming (MILP)-based planners [16, 17] have been developed that can be used to plan over a range of STL specifications but still encounter difficulties with collision avoidance when a modest number such as 5 agents are considered (Table 1).

Inspired by recent progress in learning-based planners [18, 19, 20], we propose a novel approach to planning for multi-agent systems with STL specifications that can scale beyond these limitations demonstrated on up to 32 agents. More specifically, we introduce a Graph Neural Network (GNN) based planner using Neural Ordinary Difference Equations (ODEs) [18] (Fig. 1, Sec. 4.2) trained

8th Conference on Robot Learning (CoRL 2024), Munich, Germany.

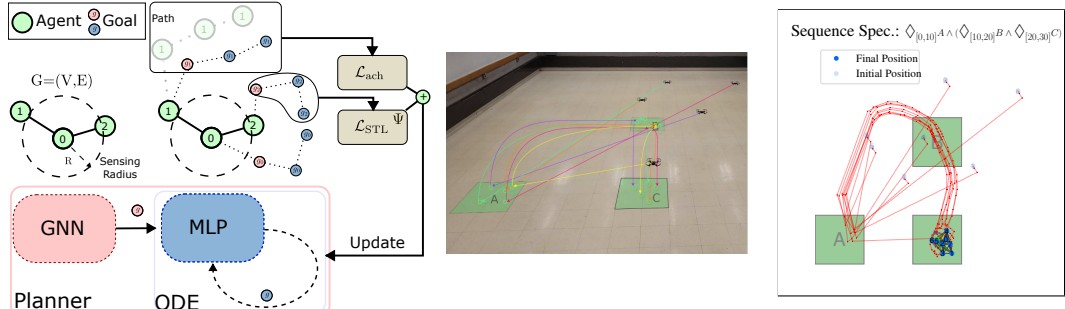

Figure 1: **(Left)** GNN-ODE Planner Architecture for Multi-Agent Systems with STL Specifications. The planner $\pi_g^{\phi_i}$ generates a sequence of goals for agent $i$ given the initial state of the system $G(0)$. The safety controller $\pi_i$ ensures that the agents do not collide while following the generated goals. A GNN encodes the graph representing the collective initial state of the system to yield an initial goal $g_i(0)$ (red) for each agent $i$. This goal $g_i(0)$ is fed into a Multi-Layer Perceptron (MLP) network to generate a new goal $g_i(1)$ (blue) which is fed back into the MLP network in a feedback loop. This is repeated for $T-1$ steps to generate a sequence of goals for the agent. The losses $\mathcal{L}_{\text{STL}}$ and $\mathcal{L}_{\text{ach}}$ are detailed in Sec. 4.1 and are used to update our planner. **(Middle)** Real world experiments on $N = 5$ drones. **(Right)** An example trajectory for $N = 8$ agents for a *seq* spec requiring agents to visit A then B and finally C in order.

end-to-end on an STL objective to generate safe and robust plans for multiple agents that can be realized using a learnable MA collision avoidance controller ([21], Sec. 4.3). To scale up, we use the ODE-based component to plan general paths that satisfy the given task while using a GNN to model agent interactions in a scalable manner to achieve coordination between the agents as they determine which ODE-generated goal trajectory to follow. Our loss components (Sec. 4.1) allow the planner to find paths that satisfy the STL objective while also being achievable in the presence of collision avoidance maneuvers and agent-to-agent interactions.

Our contributions are as follows: 1) We propose a novel scalable GNN-based planner (GNN-ODE) trained on an STL objective to generate safe and achievable plans for multiple agents. 2) We demonstrate the effectiveness of our approach on a range of STL specifications and show that our method can scale to a large number of agents and complex specifications beyond existing methods that use a state-of-the-art MILP-based planner with an average 65% improved success rate.

## 1.1 Related Work

Symbolic methods have been part of a recent resurgence as neuro-symbolic algorithms [22, 23] which aspire to combine the generalizability of neural methods with the ability of most symbolic systems to be interpretable and modifiable by human users. Notably, there have been efforts to integrate temporal logic constraints within learning-enabled controllers. In the field of Reinforcement Learning (RL), some examples of this are TLTL [24], which defines a reward function from a logic specification and reward shaping mechanisms, [25, 26] which create automata modeling a similar specification and augment RL-based algorithms used for control. This has been extended to the Multi-agent domain, which had recent work [27, 9] showing possibilities of coordinating multiple agents with diverging objectives, as well as the benefits of distributing specifications among agents in terms of scalability.

A key aspect of scaling control to higher-dimensional environments and robots involves efficiently incorporating a high-level planner. This involves decomposing complex logic planning from control tasks, allowing each component to focus on its specific role. The high-level planner focuses on logic-level planning, ensuring that the robot's actions adhere to complex specifications, such as those defined by STL. In contrast, the low-level controller acts as a tracker, executing the high-level plan accurately. Modern control methods have demonstrated this benefit as well from the burgeoning

progress in Hierarchical RL [28, 29, 30, 31, 25] methods as well as the successful integration of classical planners with advanced control schemes, including RL controllers [32].

Symbolic techniques have appeared in robot motion planning as well with the use of Signal Temporal Logic (STL) to specify objectives for multi-robot systems, which can then be solved by MILP solvers [16], graph-based algorithms [33] or sampling-based methods [34, 35]. Collision avoidance in these multi-robot systems is a challenging problem since one must also achieve the underlying objectives as well and a myriad of techniques [36, 37, 38, 21, 39] have attempted to handle this for general robot motion planning tasks. These existing methods, however, have not considered the generality of symbolic methods in specifying these objectives or quickly fail to scale as the specification dimension, robot complexity and number of agents increases.

## 2  Background

**Multi Agent Systems with Partial Observability**  We can represent a multi-agent system with $N$ agents $\{1, 2, \ldots N\}$. Each agent has its own state $s_i(t) \in \mathcal{S}_i \subset \mathcal{R}^n$, can take an action $u_i(t) \in \mathcal{U}_i \subset \mathcal{R}^m$, and the collective behavior of the agents is governed by a dynamics function $s_i(t+1) = f_i(s_i(t), u_i(t))$. For simplicity, we assume all agents have the same dynamics function $f_i = f$, state space $\mathcal{S}_i = \mathcal{S}$, and action space $\mathcal{U}_i = \mathcal{U}$. A trajectory $\tau$ is a sequence of states $\tau = (\bar{s}(0), \bar{s}(1), \ldots, \bar{s}(T_h))$ where $T_h$ is the time horizon, $\bar{s}(t) = (s_1(t), \ldots s_N(t))$, $\bar{u}(t) = (u_1(t), \ldots u_N(t))$ and a policy $\pi_i$ is a function that maps the state of agent $i$ to an action $u_i = \pi_i(s_i)$. The state of the system is partially observable, meaning that each agent can only observe its own state and the states of other agents within its sensing range.

**Signal Temporal Logic**  Signal Temporal Logic (STL) integrates both first-order logic and time-dependent modifications of linear temporal logic operators. The essential logical operators include $\wedge$ (and), $\neg$ (not), $\vee$ (or), and $\Rightarrow$ (implies). Time-dependent operators are $\Diamond_{[a,b]}$ (eventually between times $a$ and $b$), $\Box_{[a,b]}$ (globally between times $a$ and $b$), and $\mathcal{U}_{[a,b]}$ (until between times $a$ and $b$). STL formulas are defined as:

$$\phi := \mathcal{P} \mid \neg\phi \mid \phi \wedge \psi \mid \phi \vee \psi \mid \phi \Rightarrow \psi \mid \Diamond_{[a,b]}\phi \mid \Box_{[a,b]}\phi \mid \phi\,\mathcal{U}_{[a,b]}\psi,$$

where $\mathcal{P}$ is a predicate function mapping states to real values. Quantitative semantics [13, 40] of STL evaluate a robustness value, $\rho(\phi, \tau)$, which measures how strongly a state trace $\tau$ satisfies or violates $\phi$. This robustness metric is differentiable, allowing for direct optimization of STL formulas through differentiable planners like neural networks.

**Multi-agent Specification**  With regards to multi-agent systems with $N$ agents, the MA-STL specification $\Psi$ is composed from $N$ individual STL specifications $\bigwedge_{i=1}^{N} \phi_i$ where $\phi_i$ associates an STL specification for a single agent with index $i$. The MA-STL specification $\Psi$ is satisfied if all individual STL specifications are satisfied and the agents do not collide.

**Graphical Representation for Multi-Agent Systems**  Graph Neural Networks (GNNs) are adept at modeling multi-agent systems by representing agents and obstacles as vertices within a graph $G = (V, E)$. Each vertex in $V = V_a \cup V_o$ corresponds to either an agent or a static obstacle. Edges $E$ encapsulate direct interactions between vertices, with specific emphasis on agent-to-agent and agent-to-obstacle connections. We adopt a distance-based adjacency criterion where an edge $(v_i, v_j) \in E$ exists if the Euclidean distance between vertices $v_i$ and $v_j$ does not exceed a predefined threshold $R$, for capturing the local topology of agents within this range [38]. A GNN processes the graph to produce a global embedding representing the collective state of the system. This global state is further processed through specialized readout functions $r_i$, tailored to extract and map the global embedding to a specific set of actions $u_i$ for each agent [39, 21, 38].

**Barrier Certificates**  Barrier certificates [41] are a useful technique to avoid robot collisions in MA systems [42] by forcing the state of the entire system to stay within the safe region. For a state space $\mathcal{S} \subset \mathbb{R}^n$, let $\mathcal{S}_u \subset \mathcal{S}$ be the unsafe set and $\tilde{\mathcal{S}}_s = \mathcal{S} \backslash \mathcal{S}_u$ the safe set, which contains the set

of initial conditions $S_0 \subset S_s$. Also, define the space of control actions as $\mathcal{U} \subset \mathbb{R}^m$. For a dynamic system $\dot{s}(t) = f(s(t), u(t))$, a control barrier function $h : \mathbb{R}^n \mapsto \mathbb{R}$ satisfies:

$$h(s) \geq 0 \quad \forall s \in \mathcal{S}_0, \; h(s) \; < 0 \quad \forall s \in \mathcal{S}_u, \; \nabla_s h \cdot f(s, u) + \alpha(h(s)) \geq 0 \quad \forall s \in s \mid h(s) \geq 0. \tag{1}$$

For a control policy ( $\pi : \mathcal{S} \to \mathcal{U}$ ) and CBF ($h$), if ($s(0) \in s \mid h(s) \geq 0$) and the above conditions are satisfied with ($u = \pi(x)$), then ($s(t) \in s \mid h(s) \geq 0$) for all $t \in [0, \infty)$. This implies that the state never enters the unsafe set ($\mathcal{S}_u$) under $\pi$ (see [41]).

Learning-based approaches for barrier certificates [37, 39, 38, 21] have been shown to scale in the number of agents beyond existing methods for known systems. Notably, a graphical perspective of the agents and their interactions can be used to model the system in a scalable manner (Sec. 4.3).

## 3  Problem Statement

Consider a MA-STL specification $\Psi$ on $N$ agents $\mathcal{N} = \{1, 2, \ldots, N\}$, where each agent is at a position $p_i(t) \in \mathbb{P} \subset \mathbb{R}^n$ with $n$ being 2 or 3 for 2D or 3D environments respectively. Assume that each state $s_i(t)$ of agent $i$ can be directly mapped to its position $p_i(t)$, say the first $n$ elements of $s_i(t)$ by a function $\texttt{filter}_{p_i} : S \to \mathbb{P}$. Similar to Zhang et al. [21], we include a LiDAR based observation of $n_{\mathrm{rays}} > 0$ for each agent measuring the distance to the nearest obstacle in the environment with a sensing radius $R > 0$. The $j$-th ray of agent $i$ is denoted as $y_{i,j}(t)$, where $y_{i,j}(t) \in \mathbb{R}^+$ is the distance to the nearest obstacle in the direction of the $j$-th ray at time $t$.

**The MA-STL motion planning problem**  We now establish the problem of motion planning for MA-STL in multi-agent systems. Essentially, the objective is to identify a set of reference goals that when followed satisfy a given MA-STL specification, while ensuring that there are no collisions between the agents. Suppose there are $N$ agents involved, and the time bound is denoted by $T_h$. The planner $\pi_g^{\phi_i}$ generates a sequence of goals $\tau_{g_i} = (g_i(0), g_i(1), \ldots, g_i(T))$ for agent $i$ with a given plan length $T < T_h$. Each agent has a size radius represented by $r$, where $r > 0$. This means that when an agent is at position $p \in \mathcal{W}$, it is entirely contained within a ball of radius $r$ centered at $p$, denoted as $B_r(p)$. With these considerations, we can define the planning problem as follows:

**Definition 1 (Motion Planning in MA-STL)** *For a given MA-STL specification $\Psi = \bigwedge_{i=1}^N \phi_i$ and a set of $N$ agents $\mathcal{N}$, the motion planning problem is finding a distributed control policy $\pi_i$ and a planner $\pi_g^{\phi_i}$ for each agent $i$ such that the following conditions are satisfied for closed-loop trajectories of agents in $\mathcal{N}$ with length $T_h$:*

- *(Safety - Agents) For all $t \in [0, T_h]$, and for all $i, j \in \mathcal{N}$ where $i \neq j$, $||p_i(t) - p_j(t)|| \geq 2r$.*

- *(Safety - Obstacles) For all $t \in [0, T_h]$, and for all $i \in \mathcal{N}$, $y_{i,j}(t) \geq 2r$ for all $j \in [n_{rays}]$.*

- *(STL Satisfaction) There exists $t_0, t_1, \ldots, t_T$ such that $t_i \in \{0, \ldots, T_h\}$ and $t_0 < t_1 < \ldots < t_T$ such that the closed-loop trajectories $\tau = (s(t_0), s(t_1), \ldots, s(t_T))$ of the agents satisfy the MA-STL specification $\Psi$ i.e. $\rho(\Psi, \tau) \geq 0$.*

- *(Achievability) For all $i \in \mathcal{N}$, given the goal trajectory $\tau_{g_i}$ of length $T$ from $\pi_g^{\phi_i}$, the gap $D_{\tau_{g_i}}(\tau_i) = \sum_{t'=0}^T \left\| \texttt{filter}_{p_i}(s_i(t_{t'})) - \texttt{filter}_{p_i}(g_i(t')) \right\|_2 < \epsilon$ for a small $\epsilon \in \mathbb{R}^+$.*

**Scaling STL for Multi-agent Systems**
Given an MA-STL specification $\Psi$ on a system of $N$-agents we would like to provide a decentralized algorithm to execute a policy satisfying the specification with high probability. While we might assume a plan-then-execute technique [16] that finds a Piece-Wise Linear (PWL) path for each agent with $K$ segments, such an approach quickly fails to scale with specification complexity and number of agents

| N | Spec. 1 / Spec. 2 | Planning Time (s) |
|---|---|---|
| 3 | 1 seq / 2 seq | 11 / 292 |
| 5 | 1 seq / 2 seq | 211 / - |

Table 1: Planning when considering disjoint time or space [16], a PWL plan with $K = 6$ segments (1 *seq* ) / $K = 10$ segments (2 *seq*). The $X$ *seq* spec. has $X$ sequential waypoints.

when considering collisions between agents at planning time. We posit this is primarily due to its collision avoidance mechanism that introduces $\mathcal{O}(C_2^N * K^2)$ new variables, which quickly blows up (where $C_2^N = N(N-1)/2$). Consider two goal regions $A$ and $B$ and a sequential STL specification requiring agents to visit $A$ (viz. 1 *seq*) or to visit $A$ then $B$ (viz. 2 *seq*) while

avoiding collisions. Table 1 demonstrates this by timing out (over 50 minutes) for a simple STL specification with $N = 5$ agents in a 2-D environment with Single Integrator dynamics as well as all specifications and number of agents considered in this work (Sec. 5, App. B) .

Accounting for collision-avoidance independent of the objective is not novel [37, 39], but, as we argue in this paper, in order to satisfy an STL specification, one must account for the temporal nature of the specification simultaneously with performing any collision-avoidance maneuvers. An alternative, as we propose, is to plan for the objectives while adjusting for collision avoidance by means of an iterative training procedure involving the safety controller (such as GCBF+) and the planner.

## 4    Approach

Our approach integrates planning, control, and safety mechanisms in an end-to-end differentiable learning framework. We first introduce a differentiable STL framework using a neural network planner to maximize STL robustness (Sec. 4.1). For efficient multi-agent planning, we employ GNNs to model agent relationships and generate decentralized goal sequences (Sec. 4.2). To ensure collision avoidance, we define a safe set of states using GCBFs for robust control (Sec. 4.3). Finally, we discuss the training of our integrated system (Sec. 4.4).

### 4.1    Differentiable Signal Temporal Logic for Planning

Signal Temporal Logic (STL) provides a robustness metric for a given trajectory that quantifies the level of satisfaction of a specification $\phi$ defined using the STL language (Sec. 2). Consider a NN planner $\pi_g^{\phi_i}$ that takes as input the current state of the system and outputs a sequence of goals $\tau_{g_i} = (g_i(0), g_i(1), \ldots, g_i(T))$ for agent $i$ with specification $\phi_i$. We can define a loss function that attempts to maximize the STL robustness score for the specification $\phi$, given the waypoints from the planner. Prior work [19, 40] has used the differentiability of this score function to directly regularize a planner's waypoints for use by a given low-level controller $\pi_i(s_i|g_i)$ which is goal-conditioned, i.e. targeted to reach the goal $g_i$ given the current state $s_i$ of agent $i$.

For the planner architecture, similar to Xiong et al. [19], we consider using $\pi_g^{\phi_i}$ to predict the deviation between subsequent waypoints $\Delta g_i$. Based on this, to maximize the probability of satisfying the STL specification given a controller $\pi_i$, we define the loss function as:

$$
\mathcal{L}_{\pi_g^{\phi_i}, \pi_i} = \mathbb{E}_{\substack{s_i \sim S_0, \tau_{g_i} \sim \pi_g^{\phi_i}(s_i), \\ \tau_i \sim \pi_i(s_i, g_i)}} \left( \underbrace{-\lambda_{\mathrm{STL}} \rho(\phi_i, \tau_{g_i})}_{\mathcal{L}_{\mathrm{STL}}} + \underbrace{\lambda_{\mathrm{ach}} D_{\tau_{g_i}}(\tau_i)}_{\mathcal{L}_{\mathrm{ach}}} \right) \tag{2}
$$

Here we consider two loss components, the first being the STL robustness score $\rho(\phi_i, \tau_{g_i})$ of the planned waypoints $\tau_{g_i}$ and the second being the tracking error $D_{\tau_{g_i}}(\tau_i)$ of the controller $\pi_i$ with respect to the planned waypoints. The coefficients $\lambda_{\mathrm{STL}}, \lambda_{\mathrm{ach}} > 0$ are hyperparameters that control the relative importance of the two loss components ($\mathcal{L}_{\mathrm{STL}}$ and $\mathcal{L}_{\mathrm{ach}}$) in the overall loss function. The STL Loss $\mathcal{L}_{\mathrm{STL}}$ captures our objective, maximizing the STL robustness score of the planned waypoints $\tau_{g_i}$ with respect to the specification $\phi_i$. The achievable loss $\mathcal{L}_{\mathrm{ach}}$ on the other hand ensures that the controller $\pi_i$ can track the planned waypoints $\tau_{g_i}$ using a distance metric $D_{\tau_{g_i}}(\tau_i)$ (Defn. 1) that extracts the positions from $\tau_i$ using $\texttt{filter}_{p_i}$ and minimizes a normed distance between the two, i.e. $D_{\tau_{g_i}}(\tau_i) = \sum_{t=0}^{T} \| \texttt{filter}_{p_i}(s_i(kt)) - \texttt{filter}_{p_i}(g_i(t)) \|_2$ where $k > 0, k \in \mathbb{Z}^+$ is a fixed goal sampling rate during training such that $kT = T_h$. In this paper, we consider the same specification $\phi_i = \phi$ and use the same planner for all agents. This enables easy generalization to different numbers of agents during testing and allows for a more scalable approach to planning. We leave the question of how to support different specifications among agents for future work. This leads to our overall loss function for the planner and controller as $\mathcal{L}_{\pi_g^\phi, \pi} = \sum_{i=1}^{N} \mathcal{L}_{\pi_g^{\phi_i}, \pi_i}$.

### 4.2    GNNs for Planning in Multi-Agent Systems

Graphical models can be useful to scale collision avoidance in multi-agent systems [39, 21, 38] by modeling the system in a decentralized manner. Notably, by representing the agents as nodes and

their interactions as edges, we can use Graph Neural Networks (GNNs) to process a graphical view of the system as described in Sec. 2 (Fig. 1).

To handle the planning problem in multi-agent systems we describe the planner $\pi_g^\phi$. We choose a GNN-based planner that takes as input the initial state of the system $G(0)$ and outputs an initial goal $g_i(0)$ for agent $i$ taking into account the relative positions of the agents. Next we feed this goal $g_i(0)$ into a 2-layer MLP to predict the deviation $\Delta g_i$. This process is repeated for $T-1$ steps to generate a sequence of goals for the agent given the initial state. By using this GNN-based structure, we can get this sequence of goals $\tau_{g_i}$ for each agent $i$ in a single forward pass of the planner $\pi_g^\phi$.

As highlighted in Sec. 2, MA-STL can be thought of as *independent* single-agent STL specifications on the agents, albeit with an additional constraint on avoiding collisions between the agents. While collision avoidance during planning time is expensive (Sec. 3), we can attempt to plan for the objectives for a subset of the agents and use this plan with a safety scheme during run-time. Along these lines, during deployment, we use the GNN-ODE (Fig. 1) to generate a sequence of waypoints that we sequentially visit in a decentralized manner using the GCBF+ controller (Sec. 4.3). One should note this would not be straightforward if we had defined arbitrary STL specifications in the joint space of agents involving global coordination or synchronization of objectives [43, 9].

However, because this may detract from the overall objective due to collision avoidance maneuvers causing deadlocks, we update the planner iteratively by sampling the environment as detailed in Sec. 4.1 with the STL robustness score. In a sense, we "co-learn" the safety (GCBF+ controller, $\pi_i$) and objective (GNN-ODE, $\pi_g^{\phi_i}$) behavior which is a recurrent theme in recent work [19, 32] related to the safety of controllers in complex systems.

### 4.3 Collision Avoidance in MA Systems

Following [21], we define the safe set $\mathcal{S}_s \subset \mathcal{S}^N$ of an $N$-agent MAS as the set of MAS states $\bar{s}$ that satisfy the safety properties in Problem 1, i.e.,

$$\mathcal{S}_s := \left\{ \bar{s} \in \mathcal{S}^N \;\middle|\; \left( \|y_{i,j}\| > r, \; \forall i \in \mathcal{N}, \forall j \in n_{\text{rays}} \right) \bigwedge \left( \min_{i,j \in \mathcal{N}, i \neq j} \|p_i - p_j\| > 2r \right) \right\}. \tag{3}$$

Then, the unsafe, set of the MAS $\mathcal{S}_u = \mathcal{S}^N \setminus \mathcal{S}_s$ is defined as the complement of $\mathcal{S}_s$. We now define the notion of a GCBF[21]:

**Definition 2 (GCBF)** *A continuously differentiable function $h : \mathcal{S}^M \to \mathbb{R}$ is termed as a Graph CBF (GCBF) if there exists an extended class-$\mathcal{K}_\infty$ function $\alpha$ and a control policy $\pi_i : \mathcal{S}^M \to \mathcal{U}$ for each agent $i \in V_a$ of the MAS such that, for all $\bar{s} \in \mathcal{S}^N$ with $N \geq M$,*

$$\dot{h}(\bar{s}_{\mathcal{N}_i}) + \alpha(h(\bar{s}_{\mathcal{N}_i})) \geq 0, \quad \forall i \in V_a \tag{4}$$

*where for $u_j = \pi_j(\bar{s}_{\mathcal{N}_j})$ and set of neighbours $\mathcal{N}_i$ of agent $i$ in the MAS within sensing radius $R$, we have*

$$\dot{h}(\bar{s}_{\mathcal{N}_i}) = \sum_{j \in \mathcal{N}_i} \frac{\partial h(\bar{s}_{\mathcal{N}_i})}{\partial s_j} f(s_j, u_j), \tag{5}$$

From this definition, as a consequence of the results in Zhang et al. [21], if we find a control policy $\pi_i$ and GCBF $h$ such that Eq. (4) holds for all agents $i$ and all states $\bar{s} \in \mathcal{S}_s$ , then the MAS will never enter the unsafe set $\mathcal{S}_u$ under the control policy $\pi_i$.

### 4.4 End-to-End Differentiable Learning for MA-STL

By using the learning framework described in Sec. 4.2 and the safety mechanism in Sec. 4.3, we can train the planner and controller in an end-to-end differentiable manner using the loss function in Eq. (2) (Sec. 4.1). We use an iterative training loop to sample trajectories from the environment at different starting conditions and update the planner $\pi_g^\phi$ for a trained common GCBF+ controller $\pi_i = \pi$ using the loss $\mathcal{L}_{\pi_g^\phi, \pi}$.

## 5 Experiment Setup

Our experiments aim to validate the following two questions:

| Metric | Planning Time (s) ↓ | | Finish Rate (%) ↑ | | Safety Rate (%) ↑ | | Success Rate (%) ↑ | | TtR (steps) ↓ | |
|---|---|---|---|---|---|---|---|---|---|---|
| **Planner** | GNN-ODE | STLPY | GNN-ODE | STLPY | GNN-ODE | STLPY | GNN-ODE | STLPY | GNN-ODE | STLPY |
| Spec / N | | | | | | | | | | |
| Branch 8 | **0.05** | 22.48 | **100.00** | 100.00 | **100.00** | 85.00 | **100.00** | 85.00 | 712.50 | **357.00** |
| Branch 16 | **0.04** | 43.80 | **100.00** | 99.00 | **100.00** | 53.75 | **100.00** | 52.50 | 745.12 | **429.81** |
| Branch 32 | **0.03** | 87.92 | 95.00 | **96.00** | **92.50** | 20.00 | **88.12** | 18.12 | 820.90 | **572.39** |
| Cover 8 | **0.02** | 10.40 | **100.00** | 95.00 | **100.00** | 97.50 | **100.00** | 95.00 | 1062.00 | **429.76** |
| Cover 16 | **0.02** | 20.14 | **100.00** | 78.00 | **96.25** | 87.50 | **96.25** | 76.25 | 1124.25 | **536.33** |
| Cover 32 | **0.03** | 40.19 | 99.00 | 80.00 | **85.00** | 56.88 | **84.38** | 53.75 | 1252.59 | **708.22** |
| Loop 8 | **0.02** | 26.16 | **100.00** | 98.00 | **100.00** | 82.50 | **100.00** | 80.00 | 1874.00 | **1095.29** |
| Loop 16 | **0.02** | 52.79 | **100.00** | 99.00 | **97.50** | 67.50 | **97.50** | 66.25 | 1927.50 | **1301.54** |
| Loop 32 | **0.04** | 111.62 | 99.00 | **100.00** | **86.25** | 38.75 | **85.62** | 38.75 | 2110.88 | **1598.19** |
| Seq. 8 | **0.07** | 3.46 | 95.00 | **98.00** | 95.00 | **100.00** | 95.00 | **97.50** | 988.64 | **637.11** |
| Seq. 16 | **0.07** | 6.96 | **90.00** | 89.00 | **93.75** | 90.00 | **86.25** | 85.00 | 1173.44 | **785.97** |
| Seq. 32 | **0.08** | 13.62 | **89.00** | 84.00 | **76.25** | 64.38 | **66.88** | 59.38 | 1277.91 | **1013.90** |

Table 2: Performance of the two planning schemes with the number of agents ($N$) and specification complexity for the DubinsCar Environment. We note an average 65% improved success rate and highlight the best result in **bold**.

- How scalable is a neural STL planner over competing methods in terms of the number of agents and specification complexity?
- How do the distinct components of our planner (GNN and ODE) help with scalability?

To demonstrate the robustness of our method to various specifications and agent models we execute our experiments on the following robot benchmarks: 2D single integrator dynamics (App. C.1), 2D non-linear Dubins Car model (Table 2), 2D Double Integrator dynamics (App. C.3) and a real-world 3D drone quadcopter setup moving in a fixed 2D plane (App. C.4).

Our framework[1] was built using JAX [44] based off GCBF+ [21] (Sec. 4.3) with all comparisons using this underlying collision avoidance controller. To demonstrate the effectiveness of our method, we compare it against a state-of-the-art MILP-based planner (STLPY [17], Table 2) and an ablation of our planner without the GNN component (labeled ODE, Table 3).

We evaluate the planner on a range of specifications: *seq*, *cover*, *loop* and *branch*. These STL specifications can be drawn to parallels in the real-world. A *seq* task is akin to a set of drones that need to visit a series of locations in a specific order at given time intervals for logging time-sensitive information. The *cover* task depicts a scenario where each drone measures a different sensor reading but must all cover the same locations within a time interval to consolidate information. The *loop* task captures a set of surveillance drones patrolling the same areas. Lastly, consider a scenario where drones are grouped into two separate rooms with two goals present in each. Here a *branch* task could represent a common specification applied to each agent that they must visit the goals of a particular room. For a more formal description of the specifications, refer to Appendix B.

We sampled 30 random initial seeds for each experiment and report the mean planning time (in seconds), the percentage of runs in which the specification was satisfied (Finish Rate), the percentage of runs where the agent was safe (i.e. did not collide), the percentage of successful runs for each specification where the STL specification was satisfied and no collisions occurred, and time-to-reach (TtR) in number of steps (i.e. how long it took for the successful runs to complete the task).

## 6 Results

Our results (Table 2) demonstrate how differentiable STL can be used to ensure agents achieve complex objectives while avoiding collisions in multi-agent systems.

**Scalability in number of agents** We first evaluate the scalability of our approach in the number of agents ($N$) for the non-linear DubinsCar environment. From the results, we observe that the success rate decreases gradually as the number of agents increases, which is expected as the number of agents increases the complexity of the problem. The results show that the single agent view of STLPY proves unsuccessful especially in the $N = 32$ case where agent interactions are more prevalent. Notably our approach has a planning time that is 70-1000x faster than the MILP-based planner (STLPY) and does not blow up when considering a larger number of agents $N$.

---

[1]Code: https://github.com/jeappen/mastl-gcbf

| Metric | | Finish Rate (%) ↑ | | Safety Rate (%) ↑ | | Success Rate (%) ↑ | | TtR (steps) ↓ | |
|---|---|---|---|---|---|---|---|---|---|
| Spec | N | Percentage Change | ODE | Percentage Change | ODE | Percentage Change | ODE | Percentage Change | ODE |
| Branch | 8 | -2.00 | 98.00 | 0.00 | 100.00 | -2.50 | 97.50 | -26.01 | 527.21 |
| | 16 | -4.00 | 96.00 | -2.50 | 97.50 | -6.25 | 93.75 | -24.86 | 559.88 |
| | 32 | 0.00 | 95.00 | -8.78 | 84.38 | -7.08 | 81.88 | -18.18 | 671.66 |
| Cover | 8 | 0.00 | 100.00 | 0.00 | 100.00 | 0.00 | 100.00 | -28.19 | 762.57 |
| | 16 | 0.00 | 100.00 | 0.00 | 96.25 | 0.00 | 96.25 | -28.58 | 802.95 |
| | 32 | 0.00 | 99.00 | 7.35 | 91.25 | 7.40 | 90.62 | -28.48 | 895.88 |
| Loop | 8 | 0.00 | 100.00 | 0.00 | 100.00 | 0.00 | 100.00 | -16.30 | 1568.57 |
| | 16 | 0.00 | 100.00 | -11.54 | 86.25 | -11.54 | 86.25 | -17.11 | 1601.03 |
| | 32 | 0.00 | 99.00 | 0.00 | 86.25 | 0.74 | 86.25 | -13.85 | 1818.59 |
| Seq. | 8 | -26.32 | 70.00 | 5.26 | 100.00 | -26.32 | 70.00 | 31.34 | 1298.50 |
| | 16 | -32.22 | 61.00 | -1.33 | 92.50 | -28.99 | 61.25 | 15.30 | 1352.94 |
| | 32 | -47.19 | 47.00 | 26.23 | 96.25 | -28.98 | 47.50 | 7.54 | 1374.23 |

Table 3: Considering an ablation without the GNN module for the DubinsCar Environment at various scales and reporting the percentage change in values. Planning times are comparable.

**Scalability in specification complexity** For certain specifications such as *branch* and *loop*, we observe that the MILP planner computation time is significant which can add up over different agent initializations. In contrast, our planner is able to generate a solution for all the specifications quickly and consistently for different agent initial positions, motivating our learning-based approach. We further note the effect in TtR when using our algorithm. We rationalize this trade-off because our method finds longer paths that allow goals to be reached by the GCBF+ controller, which is trained to avoid other agents. This inherently reduces the number of collisions which is often a greater priority. From our ablation study (Table 3) we note the impact of the GNN module especially in terms of a 28% impact in success rate for certain specifications such as *seq* which require increased coordination among agents. We can further reason that the impact in *cover* and *loop* of the GNN module is not as great since agents are not required to reach the goals within a strict order and thus require less coordination. With regards to the lower TtR of the successful runs, the lack of a GNN module may yield plans that can satisfy the specification efficiently but fail in terms of coordination between agents (affecting the overall success rate).

## 7 Limitations

**Model-based learning** While a model-free approach to collision-avoidance [45, 32] would be more amenable to handle unknown environment dynamics, our approach is inherently model-based (as is GCBF [38, 21], MACBF [37] and CAM [39]). This is primarily due to the underlying controller and GCBF (akin to a barrier certificate), using the next state of the system while calculating the derivative for use in the loss function.

**Map Complexity, Homogeneity** Additionally, since the approach is decentralized, complex maps requiring communication and coordination between agents may cause safety issues. As mentioned in [21], it may be hard in dense regions to act in a decentralized manner thus necessitating the use of inter-agent communication. We have considered the homogeneous case in this work, where all agents have the same dynamics and STL specifications. However, in the heterogeneous case, agents may have different dynamics and STL specifications thus needing a more complex controller and a planner capable of generalizing to multiple goal positions or STL specifications. Our planner does not consider obstacles directly, although as demonstrated in the Appendix (Tables 5, 6, 7), the GCBF+ controller to an extent provides inherent collision avoidance capabilities. Finally, the approach is limited by the complexity of the environment and the number of agents. While we have shown that the approach scales well with the number of agents, the complexity of the environment and the number of obstacles may cause the planner to fail to find a achievable plan.

## 8 Conclusion

In this work, we have presented a novel approach to planning for multi-agent systems with Signal Temporal Logic specifications. Primarily we have shown that by using a differentiable STL robustness metric, we can optimize for the satisfaction of complex temporal specifications given a controller with MA collision avoidance capabilities. We demonstrate that by training a GNN-ODE planner with a carefully constructed loss function we can overcome the limitations of the plan-then-execute approach and scale to complex specifications and large numbers of agents.

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

# Contents

# A  Implementation Details

**Node features and edge features**    Following Zhang et al. [21], the node features $v_i \in \mathbb{R}^{\rho_v}$ encode information specific to each node in our graph observation. Here, we set $\rho_v = 3$ and use the node features $v_i$ to one-hot encode the type of the node as either an agent node, goal node or LiDAR ray hitting point node. The edge features $e_{ij} \in \mathbb{R}^{\rho_e}$, where $\rho_e > 0$ is the edge dimension, are defined as the information shared from node $j$ to the agent at node $i$, which depends on the states of the nodes $i$ and $j$. Since the safety objective depends on the relative positions, one component of the edge features is $p_{ij} = p_j - p_i$. The remaining edge features can be set depending on system dynamics, such as, relative velocities for double integrator dynamics.

**Computation Resources**    All training procedures were ran on an AWS `g4dn.xlarge` instance or equivalent with 4 Intel Xeon-based CPU Cores and 16 GB of RAM with an Nvidia T4 GPU.

**Evaluation Details**    Since we consider objectives that require agents to navigate close to one another at/near termination subsequently blocking the goal locations (A,B,C,D in Table 4), safety rates were reported until the point an agent had completed their plan. This can be thought of as an alternative to the agents navigating to a 'safe' position upon completing their specification/plan. In a drone setting, we captured this behavior by landing the drones at an agent-specific location upon completing their specification.

**Planner Details**    For all plans, at any time step $t$, planning step $t'$, each agent $i$ proceeded to the next waypoint $g_i(t' + 1)$ only when they reached goal $g_i(t')$ within some threshold distance $r_{goal} = 0.3$ at a time $t \geq k(t' + 1)$ where $k$ is the goal sampling interval (Sec. 4.1). This allowed all agents to reach the waypoints in the plan without a strict time restriction on the plan duration. The asynchronous nature of our plans (among agents) fits our problem description (Defn. 1), specifically the STL Satisfaction criteria. We leave the setting where agents follow a synchronized plan to future work. For a given plan of length $T$ with goal sample interval $k$ (values in Table 4), the maximum trajectory length horizon during evaluation $T_h$ was $5kT$.

## A.1  Environment Details

Here, we provide the details of each experiment environment as taken from Zhang et al. [21]. We used a common simulation time step $\delta t = 0.03$ across all three environments.

**SingleIntegrator**    We use single integrator dynamics as the base environment to verify the correctness of the implementation and to show the performance of the methods when there are no control input limits. The dynamics is given as $\dot{x}_i = v_i$, where $x_i = [p_i^x, p_i^y]^\top \in \mathbb{R}^2$ is the position of the $i$-th agent and $v_i = [v_i^x, v_i^y]^\top$ its velocity. In this environment, we use $e_{ij} = x_j - x_i$ as the edge information.

**DoubleIntegrator**    We use double integrator dynamics for this environment. The state of agent $i$ is given by $x_i = [p_i^x, p_i^y, v_i^x, v_i^y]^\top$, where $[p_i^x, p_i^y]^\top$ is the position of the agent, and $[v_i^x, v_i^y]^\top$ is the velocity. The action of agent $i$ is given by $u_i = [a_i^x, a_i^y]^\top$, i.e., the acceleration. The dynamics function is given by:

$$\dot{x}_i = [v_i^x, v_i^y, a_i^x, a_i^y]^\top \tag{6}$$

In this environment, we use $e_{ij} = x_j - x_i$ as the edge information.

**DubinsCar**    We use the standard Dubin's car model in this environment. The state of agent $i$ is given by $x_i = [p_i^x, p_i^y, \theta_i, v_i]^\top$, where $[p_i^x, p_i^y]^\top$ is the position of the agent, $\theta_i$ is the heading, and $v_i$ is the speed. The action of agent $i$ is given by $u_i = [\omega_i, a_i]^\top$ containing angular velocity and acceleration magnitude. The dynamics function is given by:

$$\dot{x}_i = [v_i \cos(\theta_i), v_i \sin(\theta_i), \omega_i, a_i]^\top \tag{7}$$

We use $e_{ij} = e_j(x_j) - e_i(x_i)$ as the edge information, where $e_i(x_i) = [p_i^x, p_i^y, v_i \cos(\theta_i), v_i \sin(\theta_i)]^\top$.

## B  STL Specifications

We formally define the Signal Temporal Logic (STL) specifications used in the experiments in Table 4. The specifications include a sequential waypoint task (*seq*), a coverage task (*cover*), a loop task (*loop*), and a branching task (*branch*). The specifications are defined over a time horizon $T$ and are satisfied if the agents satisfy the corresponding STL formula. We use four markers $A$, $B$, $C$, and $D$ to represent rectangular predicates centered around x-y coordinates $[0,0]$, $[2,2]$, $[2,0]$, and $[0,2]$, respectively. The predicates are defined as $p_i = \text{dist}(s_i, p_i) \leq 1.0$ where $\text{dist}(s_i, p_i)$ is the L1-norm $(|\cdot|_1)$ distance between the agent $i$'s state $s_i$ and the predicate $p_i$.

| Spec. | Description | Formula | T | k |
|:---:|:---:|:---:|:---:|:---:|
| *seq* | Sequence of goals | $\Diamond_{[0,T/3]}(A) \wedge \Diamond_{[T/3,2T/3]}(B) \wedge \Diamond_{[2T/3,T]}(C)$ | 15 | 20 |
| *cover* | Coverage over goals | $\Diamond_{[0,T]}(A) \wedge \Diamond_{[0,T]}(B) \wedge \Diamond_{[0,T]}(C)$ | 15 | 20 |
| *loop* | Loop over goals | $\Box_{[0,T/2]} \left( \Diamond_{[0,T/2]}(A) \wedge \Diamond_{[0,T/2]}(B) \wedge \Diamond_{[0,T/2]}(C) \right)$ | 30 | 20 |
| *signal* | Loop then move to final | $(loop U_{[0,1]} \Psi_1) \wedge \Diamond_{[0,T]}(D)$ | 30 | 20 |
| *branch* | Branching | $\left( \Diamond_{[0,T]}(A) \wedge \Diamond_{[0,T]}(B) \right) \vee \left( \Diamond_{[0,T]}(C) \wedge \Diamond_{[0,T]}(D) \right)$ | 20 | 10 |

Table 4: STL specifications used in the experiments. $T$ and $k$ are the specification lengths and goal sample intervals respectively.

## C  Additional Environments (including Obstacles)

In Tables 5, 6 and 7 we show results for the various environments and obstacle scenarios. While our GNN-ODE has an initial GNN module which can observe these obstacles, and is also trained to generate initial goals that are 'achievable', the GNN-ODE is inherently limited to only consider obstacles within the sensing radius $R$ of the agents at planning time (i.e. $t = 0$). As in Zhang et al. [21], the GCBF+ controller is trained to avoid obstacles. Thus, with a robust plan, we can achieve reasonably high success rates in this setting as well due to the run-time collision avoidance maneuvers. Planning times are nearly similar to the results in Sec. 6 (Table 2) likely because our learning-based planners do not use environment dynamics at inference time and should have a similar computation cost after training is complete. For this reason, to avoid clutter, we omit this column in the following tables. We include results from the ODE ablation of our method as shown in Table 3 under the column 'ODE'. Additional simulation videos are hosted online[2].

### C.1  SingleIntegrator Environment

In Table 5 we contain the results for various combinations of specifications, 8 sampled obstacle positions (marked 'Y' if present, 'N' otherwise), and number of agents in the SingleIntegrator Environment. We observe that the GNN-ODE planner outperforms the other planners in terms of planning time and success rate across all the specifications and obstacles. We note the average improvement in success rate of 10% for our GNN-ODE planner over the MILP planner which is not as large as the improvement in the non-linear DubinsCar environment (Table 2, 6). This is due to the SingleIntegrator environment being less constrained and the MILP planner being able to find a feasible solution more easily.

### C.2  DubinsCar Environment

In Table 6 we contain the results for various combinations of specifications, 8 sampled obstacle positions (marked 'Y' if present, 'N' otherwise), and number of agents in the DubinsCar Environment. On average, with obstacles present as well we get a 69% improvement in success rate for our GNN-ODE planner over the MILP planner, primarily due to the non-linear dynamics of the DubinsCar environment being challenging for the collision avoidance controller.

---

[2]Site: https://jeappen.github.io/mastl-gcbf-website/

| Metric | | | Finish Rate ↑ | | | Safety Rate ↑ | | | Success Rate ↑ | | | TtR ↓ | | |
|---|---|---|---|---|---|---|---|---|---|---|---|---|---|---|
| Planner | | | GNN-ODE | ODE | STLPY | GNN-ODE | ODE | STLPY | GNN-ODE | ODE | STLPY | GNN-ODE | ODE | STLPY |
| Spec | Obs | N | | | | | | | | | | | | |
| Branch | N | 8 | 100.00 | 100.00 | 100.00 | 100.00 | 100.00 | 100.00 | 100.00 | 100.00 | 100.00 | 1000.75 | 396.50 | 257.25 |
| | | 16 | 100.00 | 100.00 | 100.00 | 100.00 | 100.00 | 100.00 | 100.00 | 100.00 | 100.00 | 1214.50 | 443.00 | 280.87 |
| | | 32 | 98.00 | 100.00 | 99.00 | 100.00 | 100.00 | 98.75 | 97.50 | 100.00 | 97.50 | 664.71 | 1005.81 | 320.23 |
| | Y | 8 | 100.00 | 93.00 | 98.00 | 100.00 | 100.00 | 95.00 | 100.00 | 92.50 | 92.50 | 1015.75 | 408.95 | 292.57 |
| | | 16 | 99.00 | 96.00 | 94.00 | 97.50 | 100.00 | 95.00 | 96.25 | 96.25 | 88.75 | 1168.80 | 471.30 | 314.99 |
| | | 32 | 97.00 | 98.00 | 94.00 | 98.12 | 100.00 | 92.50 | 95.00 | 97.50 | 86.88 | 1812.59 | 1018.77 | 356.09 |
| Cover | N | 8 | 98.00 | 100.00 | 95.00 | 100.00 | 100.00 | 100.00 | 97.50 | 100.00 | 95.00 | 884.86 | 653.00 | 342.93 |
| | | 16 | 99.00 | 100.00 | 90.00 | 100.00 | 100.00 | 100.00 | 98.75 | 100.00 | 90.00 | 1024.40 | 758.25 | 364.56 |
| | | 32 | 98.00 | 98.00 | 96.00 | 100.00 | 100.00 | 97.50 | 98.12 | 97.50 | 93.12 | 1409.89 | 1237.41 | 447.66 |
| | Y | 8 | 95.00 | 95.00 | 88.00 | 100.00 | 100.00 | 97.50 | 95.00 | 95.00 | 87.50 | 1068.46 | 744.83 | 356.17 |
| | | 16 | 96.00 | 98.00 | 91.00 | 100.00 | 100.00 | 97.50 | 96.25 | 97.50 | 91.25 | 1040.65 | 1043.66 | 386.12 |
| | | 32 | 96.00 | 98.00 | 91.00 | 100.00 | 99.38 | 96.88 | 96.25 | 97.50 | 88.75 | 1491.06 | 1297.69 | 488.79 |
| Loop | N | 8 | 100.00 | 100.00 | 100.00 | 100.00 | 100.00 | 100.00 | 100.00 | 100.00 | 100.00 | 1890.25 | 2506.00 | 751.00 |
| | | 16 | 100.00 | 100.00 | 100.00 | 100.00 | 100.00 | 100.00 | 100.00 | 100.00 | 100.00 | 1445.75 | 2120.12 | 855.38 |
| | | 32 | 100.00 | 100.00 | 100.00 | 100.00 | 100.00 | 100.00 | 100.00 | 100.00 | 100.00 | 2332.19 | 2635.75 | 1062.25 |
| | Y | 8 | 100.00 | 100.00 | 90.00 | 95.00 | 100.00 | 92.50 | 95.00 | 100.00 | 82.50 | 1434.00 | 2112.25 | 841.32 |
| | | 16 | 99.00 | 96.00 | 93.00 | 96.25 | 100.00 | 96.25 | 95.00 | 96.25 | 88.75 | 2091.53 | 2315.44 | 927.83 |
| | | 32 | 99.00 | 99.00 | 96.00 | 96.25 | 99.38 | 96.25 | 95.00 | 98.75 | 91.88 | 2453.19 | 2769.78 | 1137.71 |
| Sequence | N | 8 | 100.00 | 53.00 | 90.00 | 100.00 | 100.00 | 100.00 | 100.00 | 52.50 | 90.00 | 905.50 | 1072.30 | 434.44 |
| | | 16 | 99.00 | 41.00 | 73.00 | 100.00 | 100.00 | 100.00 | 98.75 | 41.25 | 72.50 | 761.08 | 1119.07 | 470.17 |
| | | 32 | 98.00 | 33.00 | 66.00 | 100.00 | 100.00 | 100.00 | 98.12 | 33.12 | 65.62 | 897.41 | 1464.39 | 661.21 |
| | Y | 8 | 98.00 | 45.00 | 88.00 | 100.00 | 100.00 | 100.00 | 97.50 | 45.00 | 87.50 | 731.43 | 1239.50 | 470.39 |
| | | 16 | 98.00 | 36.00 | 62.00 | 100.00 | 100.00 | 100.00 | 97.50 | 36.25 | 62.50 | 1030.83 | 1272.50 | 548.39 |
| | | 32 | 98.00 | 28.00 | 74.00 | 100.00 | 100.00 | 99.38 | 98.12 | 28.12 | 73.12 | 1186.13 | 1649.19 | 797.86 |

Table 5: Performance of different planner modules with the scalability in the number of agents ($N$) and specification complexity for the SingleIntegrator Environment.

| Metric | | | Finish Rate ↑ | | | Safety Rate ↑ | | | Success Rate ↑ | | | TtR ↓ | | |
|---|---|---|---|---|---|---|---|---|---|---|---|---|---|---|
| Planner | | | GNN-ODE | ODE | STLPY | GNN-ODE | ODE | STLPY | GNN-ODE | ODE | STLPY | GNN-ODE | ODE | STLPY |
| Spec | Obs | N | | | | | | | | | | | | |
| Branch | N | 8 | 100.00 | 98.00 | 100.00 | 100.00 | 100.00 | 85.00 | 100.00 | 97.50 | 85.00 | 1768.75 | 525.57 | 357.00 |
| | | 16 | 100.00 | 96.00 | 99.00 | 100.00 | 97.50 | 53.75 | 100.00 | 93.75 | 52.50 | 1856.12 | 565.09 | 429.81 |
| | | 32 | 95.00 | 94.00 | 96.00 | 92.50 | 86.25 | 20.00 | 88.12 | 82.50 | 18.12 | 820.90 | 674.52 | 572.39 |
| | Y | 8 | 100.00 | 100.00 | 95.00 | 97.50 | 97.50 | 67.50 | 97.50 | 97.50 | 62.50 | 728.50 | 562.79 | 373.89 |
| | | 16 | 99.00 | 95.00 | 95.00 | 92.50 | 90.00 | 47.50 | 91.25 | 86.25 | 45.00 | 1828.95 | 595.29 | 474.16 |
| | | 32 | 95.00 | 85.00 | 90.00 | 86.88 | 74.38 | 32.50 | 81.88 | 63.75 | 25.00 | 841.66 | 708.91 | 586.89 |
| Cover | N | 8 | 100.00 | 100.00 | 95.00 | 100.00 | 100.00 | 97.50 | 100.00 | 100.00 | 95.00 | 1062.00 | 754.57 | 429.76 |
| | | 16 | 100.00 | 100.00 | 78.00 | 96.25 | 97.50 | 87.50 | 96.25 | 97.50 | 76.25 | 1127.00 | 802.70 | 536.33 |
| | | 32 | 99.00 | 99.00 | 80.00 | 85.00 | 92.50 | 56.88 | 84.38 | 91.88 | 53.75 | 1252.59 | 883.31 | 708.22 |
| | Y | 8 | 98.00 | 98.00 | 93.00 | 97.50 | 95.00 | 92.50 | 95.00 | 92.50 | 85.00 | 1094.07 | 821.71 | 460.30 |
| | | 16 | 96.00 | 93.00 | 85.00 | 98.75 | 92.50 | 76.25 | 95.00 | 85.00 | 67.50 | 1135.61 | 867.24 | 571.55 |
| | | 32 | 93.00 | 93.00 | 78.00 | 81.25 | 83.12 | 53.75 | 75.62 | 76.88 | 49.38 | 1251.20 | 972.33 | 674.09 |
| Loop | N | 8 | 100.00 | 100.00 | 98.00 | 100.00 | 100.00 | 82.50 | 100.00 | 100.00 | 80.00 | 1874.00 | 1570.07 | 1092.79 |
| | | 16 | 100.00 | 100.00 | 100.00 | 100.00 | 86.25 | 76.25 | 100.00 | 86.25 | 76.25 | 1963.12 | 1601.03 | 1251.62 |
| | | 32 | 98.00 | 99.00 | 100.00 | 100.00 | 86.25 | 38.75 | 97.50 | 86.25 | 38.75 | 1936.27 | 1818.59 | 1598.19 |
| | Y | 8 | 95.00 | 88.00 | 78.00 | 100.00 | 60.00 | 92.50 | 95.00 | 55.00 | 70.00 | 1894.33 | 4554.25 | 1310.58 |
| | | 16 | 96.00 | 91.00 | 90.00 | 90.00 | 28.75 | 66.25 | 88.75 | 22.50 | 57.50 | 1969.37 | 4481.38 | 1378.32 |
| | | 32 | 89.00 | 91.00 | 88.00 | 80.62 | 13.75 | 31.25 | 76.25 | 8.12 | 24.38 | 2138.35 | 4274.29 | 1672.91 |
| Sequence | N | 8 | 98.00 | 70.00 | 98.00 | 97.50 | 100.00 | 100.00 | 95.00 | 70.00 | 97.50 | 1246.43 | 1298.50 | 637.11 |
| | | 16 | 95.00 | 70.00 | 89.00 | 96.25 | 100.00 | 90.00 | 92.50 | 70.00 | 85.00 | 1188.14 | 1577.48 | 785.97 |
| | | 32 | 89.00 | 70.00 | 84.00 | 76.25 | 77.50 | 64.38 | 66.88 | 63.75 | 59.38 | 1277.91 | 1715.05 | 1013.90 |
| | Y | 8 | 95.00 | 62.00 | 80.00 | 100.00 | 92.50 | 95.00 | 95.00 | 57.50 | 80.00 | 1572.68 | 1537.60 | 671.38 |
| | | 16 | 89.00 | 60.00 | 75.00 | 86.25 | 75.00 | 81.25 | 78.75 | 50.00 | 70.00 | 1175.86 | 1640.81 | 839.37 |
| | | 32 | 78.00 | 62.00 | 68.00 | 77.50 | 53.75 | 53.75 | 61.25 | 39.38 | 41.88 | 1293.39 | 1802.09 | 1042.54 |

Table 6: Performance of different planner modules with the scalability in the number of agents ($N$) and specification complexity for the DubinsCar Environment with obstacles.

## C.3 DoubleIntegrator Environment

In Table 7 we contain the results for various combinations of specifications, 8 sampled obstacle (Obs) positions (marked 'Y' if present, 'N' otherwise), and number of agents (N) in the DoubleIntegrator Environment. The average improvement in success rate of 11% for our GNN-ODE planner over the MILP planner is similar to the SingleIntegrator environment (Table 5) due to GCBF+controller being more effective at collision avoidance with the linear dynamics of the DoubleIntegrator environment.

## C.4 Real-world Drone Experiments

The experimental validation of this methodology involved deploying a fleet of 5 DJI Tello Ryze drones to track the trajectories generated via the Dubins Car model. The drones were configured in

| Metric | | | Finish Rate ↑ | | | Safety Rate ↑ | | | Success Rate ↑ | | | TtR ↓ | | |
|---|---|---|---|---|---|---|---|---|---|---|---|---|---|---|
| Planner | | | GNN-ODE | ODE | STLPY | GNN-ODE | ODE | STLPY | GNN-ODE | ODE | STLPY | GNN-ODE | ODE | STLPY |
| Spec | Obs | N | | | | | | | | | | | | |
| Branch | N | 8 | 100.00 | 100.00 | 100.00 | 100.00 | 100.00 | 100.00 | 100.00 | 100.00 | 100.00 | 1384.75 | 536.00 | 379.50 |
| | | 16 | 100.00 | 100.00 | 91.00 | 100.00 | 100.00 | 100.00 | 100.00 | 100.00 | 91.25 | 1768.38 | 1577.50 | 474.44 |
| | | 32 | 99.00 | 91.00 | 73.00 | 100.00 | 100.00 | 100.00 | 99.38 | 90.62 | 72.50 | 2510.85 | 2206.44 | 617.02 |
| | Y | 8 | 100.00 | 100.00 | 98.00 | 97.50 | 100.00 | 100.00 | 97.50 | 100.00 | 97.50 | 2774.50 | 533.50 | 390.57 |
| | | 16 | 99.00 | 99.00 | 94.00 | 98.75 | 98.75 | 100.00 | 97.50 | 97.50 | 93.75 | 2923.95 | 1594.28 | 533.49 |
| | | 32 | 99.00 | 91.00 | 68.00 | 96.25 | 93.75 | 98.12 | 95.00 | 85.62 | 67.50 | 2543.79 | 2263.78 | 666.90 |
| Cover | N | 8 | 100.00 | 100.00 | 93.00 | 100.00 | 100.00 | 100.00 | 100.00 | 100.00 | 92.50 | 1681.00 | 738.00 | 572.20 |
| | | 16 | 100.00 | 100.00 | 89.00 | 100.00 | 100.00 | 100.00 | 100.00 | 100.00 | 88.75 | 2127.43 | 894.50 | 645.93 |
| | | 32 | 96.00 | 75.00 | 76.00 | 100.00 | 100.00 | 100.00 | 95.62 | 75.00 | 76.25 | 2201.11 | 1542.29 | 877.32 |
| | Y | 8 | 100.00 | 100.00 | 90.00 | 100.00 | 100.00 | 97.50 | 100.00 | 100.00 | 87.50 | 1649.50 | 767.50 | 498.07 |
| | | 16 | 99.00 | 100.00 | 89.00 | 98.75 | 98.75 | 100.00 | 97.50 | 98.75 | 88.75 | 2123.30 | 926.75 | 756.73 |
| | | 32 | 95.00 | 79.00 | 78.00 | 95.00 | 96.25 | 98.75 | 90.62 | 76.25 | 77.50 | 1630.34 | 1625.60 | 949.62 |
| Loop | N | 8 | 95.00 | 98.00 | 100.00 | 100.00 | 100.00 | 100.00 | 95.00 | 97.50 | 100.00 | 1951.71 | 2716.54 | 1219.75 |
| | | 16 | 98.00 | 99.00 | 100.00 | 100.00 | 100.00 | 100.00 | 97.50 | 98.75 | 100.00 | 2612.70 | 3273.04 | 1781.62 |
| | | 32 | 98.00 | 93.00 | 96.00 | 100.00 | 100.00 | 100.00 | 98.12 | 93.12 | 96.25 | 3271.47 | 5260.64 | 2703.45 |
| | Y | 8 | 95.00 | 98.00 | 100.00 | 100.00 | 95.00 | 100.00 | 95.00 | 92.50 | 100.00 | 2533.32 | 2785.64 | 1229.75 |
| | | 16 | 96.00 | 99.00 | 98.00 | 100.00 | 96.25 | 97.50 | 96.25 | 95.00 | 95.00 | 2713.48 | 3431.33 | 1830.21 |
| | | 32 | 98.00 | 93.00 | 97.00 | 98.75 | 80.62 | 93.75 | 96.25 | 75.62 | 90.62 | 3353.94 | 5251.58 | 2850.15 |
| Sequence | N | 8 | 100.00 | 80.00 | 85.00 | 100.00 | 100.00 | 100.00 | 100.00 | 80.00 | 85.00 | 1565.50 | 1162.35 | 693.67 |
| | | 16 | 99.00 | 88.00 | 70.00 | 100.00 | 100.00 | 100.00 | 98.75 | 87.50 | 70.00 | 1667.12 | 1472.54 | 926.60 |
| | | 32 | 81.00 | 49.00 | 28.00 | 100.00 | 100.00 | 100.00 | 80.62 | 48.75 | 28.12 | 1929.73 | 1979.83 | 912.03 |
| | Y | 8 | 100.00 | 88.00 | 80.00 | 100.00 | 100.00 | 100.00 | 100.00 | 87.50 | 80.00 | 1552.75 | 1380.57 | 846.60 |
| | | 16 | 100.00 | 84.00 | 60.00 | 100.00 | 98.75 | 97.50 | 100.00 | 82.50 | 60.00 | 1684.75 | 1574.84 | 960.90 |
| | | 32 | 84.00 | 51.00 | 34.00 | 99.38 | 93.12 | 93.75 | 83.12 | 49.38 | 33.12 | 1969.68 | 2003.38 | 930.35 |

Table 7: Performance of different planner modules with the scalability in the number of agents ($N$) and specification complexity for the DoubleIntegrator Environment.

WiFi mode to enable swarm behavior which was facilitated through the open-source DJITelloPy [3] library.

Each Tello drone is equipped with an Inertial Measurement Unit (IMU), a forward-facing camera, and a downward-facing camera. The latter is useful for precise hovering and position estimation using the Vision Positioning System (VPS). However, this system is inaccurate and unreliable as the drones do not possess other sensors like lidar or depth cameras. To mitigate drift and correct the position estimate errors, ArUco tags were utilized to make the trajectory following robust for each drone. This ensured the swarm of drones could accurately follow the designated trajectory as evidenced in the simulation results.

# D   Additional Baselines

We focus on the most complex non-linear environment as presented in the main paper (DubinsCar) and have presented the results over 30 initial seeds in Table 8 and 9 and included bar charts with error bars for the main experiments in Figures 2 and 3. We chose this format as including all values in the tables created excessive clutter.

## D.1   Following a single common plan

Since the tasks are homogenous, one might attempt to follow a common plan among agents with a formation. To compare this, we include results on a common shared plan generated by the STLPY solver [17] (marked STLPY (S)) for the specifications considered in Table 8 and 9. The initial starting state of this plan was near the first predicate in each specification (viz. goal $A$).

An obvious benefit of a single plan generated by the STLPY solver is a reduced planning time. Using a single common plan also enforced a rudimentary level of coordination among agents as evident in the *branch* task reducing the number of crossing paths (present in the earlier individually generated STLPY plans). However, in long horizon specifications such as *loop* and *signal* this effect is detrimental causing increased safety violations (leading to a low overall success rate) due to a lack of a more informed coordination among agents. We posit the fixed predicate sizes further play a role in preventing this coordination scheme to succeed.

---

[3] https://github.com/damiafuentes/DJITelloPy

| | | Planning Time (s) ↓ | | | | | Success Rate ↑ | | | | | TtR ↓ | | | | |
|---|---|---|---|---|---|---|---|---|---|---|---|---|---|---|---|---|
| Planner Spec | N | CE [46] | GNN-ODE | ODE | STLPY | STLPY (S) | CE [46] | GNN-ODE | ODE | STLPY | STLPY (S) | CE [46] | GNN-ODE | ODE | STLPY | STLPY (S) |
| Branch | 8 | 8.29 | **0.01** | **0.01** | 22.48 | 1.53 | 87.50 | **100.00** | 98.75 | 85.00 | **100.00** | 598.21 | 708.25 | 518.89 | **357.00** | 520.00 |
| | 16 | 16.73 | **0.02** | **0.02** | 43.80 | 1.55 | 80.00 | **100.00** | 96.67 | 52.50 | 99.17 | 638.81 | 740.77 | 556.64 | **429.81** | 582.44 |
| | 32 | 35.62 | **0.03** | **0.03** | 87.92 | 1.52 | 65.62 | **85.10** | 81.35 | 18.12 | 83.23 | 757.52 | 823.10 | 651.32 | **572.39** | 695.18 |
| Cover | 8 | 16.20 | 0.02 | **0.01** | 10.40 | 2.56 | 95.00 | **100.00** | 98.75 | 95.00 | **100.00** | 1044.00 | 1056.25 | 758.04 | **429.76** | 845.00 |
| | 16 | 30.73 | **0.02** | **0.02** | 20.14 | 2.58 | 76.25 | **97.50** | 97.08 | 76.25 | 87.71 | 1159.90 | 1113.79 | 795.49 | **536.33** | 989.38 |
| | 32 | 63.98 | 0.03 | **0.02** | 40.19 | 2.61 | 52.50 | 81.77 | **89.38** | 53.75 | 54.90 | 1319.63 | 1241.27 | 893.52 | **708.22** | 1218.61 |
| Loop | 8 | 23.41 | 0.02 | **0.01** | 26.16 | 4.37 | 12.50 | **99.58** | 97.50 | 80.00 | 85.83 | 1757.25 | 1762.04 | 4192.00 | **1095.29** | 1179.33 |
| | 16 | 52.27 | **0.02** | **0.02** | 52.79 | 4.38 | 7.50 | **98.96** | 96.25 | 66.25 | 59.38 | 2884.33 | 1819.61 | 4139.50 | **1301.54** | 1379.57 |
| | 32 | 107.17 | **0.03** | **0.03** | 111.62 | 4.41 | 6.25 | **97.60** | 72.50 | 38.75 | 26.35 | 3111.50 | 1951.11 | 4478.88 | **1598.19** | 1940.58 |
| Sequence | 8 | 16.35 | 0.02 | **0.01** | 3.46 | 0.66 | 95.00 | 97.92 | 77.50 | 97.50 | **100.00** | 1044.00 | 1069.44 | 1296.13 | **637.11** | 850.75 |
| | 16 | 29.80 | **0.02** | **0.02** | 6.96 | 0.68 | 80.00 | 89.38 | 72.50 | 85.00 | **94.58** | 1156.38 | 1169.49 | 1337.42 | **785.97** | 957.92 |
| | 32 | 63.99 | **0.03** | **0.03** | 13.62 | 0.68 | 48.12 | 61.88 | 49.58 | 59.38 | **63.23** | 1307.78 | 1226.14 | 1376.86 | **1013.90** | 1234.37 |
| Signal | 8 | 56.59 | 0.02 | **0.01** | 13.60 | 17.87 | 0.00 | **100.00** | **100.00** | 97.92 | **100.00** | - | 2338.50 | 4179.25 | **976.53** | 1137.25 |
| | 16 | 110.95 | **0.02** | **0.02** | 23.97 | 18.21 | 0.00 | **100.00** | 91.25 | 91.88 | 93.33 | - | 2352.67 | 4314.25 | **1116.07** | 1331.53 |
| | 32 | 228.49 | **0.03** | **0.03** | 47.99 | 18.25 | 0.62 | **76.25** | 67.50 | 50.31 | 47.81 | 1761.00 | 2479.90 | 4664.87 | **1290.72** | 1831.21 |

Table 8: Performance of **STLPY** [17]: (multi-agent), **STLPY(S)** [17] : STLPY with a single common plan, **CE** [46] : Centralized Counterexample guided planner and our approaches (**ODE**, **GNN-ODE**). vs the number of agents ($N$) and specification complexity for the DubinsCar Environment. The results are averaged over 30 seeds and we highlight the best result in **bold**.

## D.2  Centralized Planner

We found that the challenges in scalable collision avoidance modeling with the PWL MA-STL planner [16] (discussed in Section 3) were also present in other centralized STL planning strategies [46]. Namely, directly adding predicates for collision avoidance into the STL specification proved intractable for optimization over the long planning horizons considered due to an $\mathcal{O}(C_2^N * K^2)$ variable blowup for $N$ agents and planning horizon $K$. To address this following our proposed method, we attempt planning for the joint multi-agent task without considering collisions during planning, while introducing run-time collision avoidance schemes like GCBF+. As a result, in the simulation step of [46] (Alg. 1, line 4), the GCBF+policy would be incorporated into the rollout. However, in our experiments, we found that the gradient vanishes too quickly (within 50 time steps) to meaningfully differentiate through the trajectory, which spans over 200 time steps for all our tasks (Table 4, Appendix B). Consequently, the algorithm in [46] cannot directly handle the longer-range STL tasks considered in our work. Additionally, we noted that the proposed approach was not robust to many initial starting positions, often leading to unfruitful plans.

To build a centralized planner with [46], we opted to run the proposed method without incorporating collision avoidance during planning time simulations. This choice allowed us to achieve longer optimization horizons within a limited initial starting position range. The resulting plans were executed using the trained GCBF+ controller. The results, as shown marked CE in 8 and 9 for the DubinsCar Environment, indicate that the proposed approach performed demonstrably worse than ours and the other compared methods (note Success Rate). The planning time of this centralized approach scaled linearly with the number of agents since agent-agent interactions were not considered. These results further point to the importance of an achievability loss ($\mathcal{L}_{ach}$, Sec. 4.1) when planning in the presence of run-time collision avoidance. This outcome reinforces the notion that current methods for centralized planning in multi-agent STL requires further advancements to effectively handle scalable collision avoidance.

## D.3  Signal Specification (demonstrating Until, $U$)

In Table 8 we include a *signal* specification demonstrating the Until operator. This is a variant of our *loop* specification (Table 4, Appendix B) called *signal* with an additional predicate $\Psi_1$ representing reaching goal $A$ twice. If *loop* is represented as $\Phi_1$, and $\Psi_2$ represents reaching a new goal $D$, the *signal* specification is $(\Phi_1 U_{[0,1]} \Psi_1) \wedge \Psi_2$ i.e. *loop* $A$, $B$, and $C$ until $A$ is reached twice, then reach $D$.

| Planner |  | Finish Rate ↑ | | | | | Safety Rate ↑ | | | | |
|---|---|---|---|---|---|---|---|---|---|---|---|
| Spec | N | CE [46] | GNN-ODE | ODE | STLPY | STLPY (S) | CE [46] | GNN-ODE | ODE | STLPY | STLPY (S) |
| Branch | 8 | 88.00 | **100.00** | 99.00 | **100.00** | **100.00** | **100.00** | **100.00** | **100.00** | 85.00 | **100.00** |
|  | 16 | 80.00 | **100.00** | 98.00 | 99.00 | **100.00** | 97.50 | **100.00** | 98.54 | 53.75 | 99.58 |
|  | 32 | 76.00 | **98.00** | 94.00 | 90.00 | 95.00 | **88.75** | 87.19 | 85.21 | 32.50 | 86.88 |
| Cover | 8 | **100.00** | **100.00** | 99.00 | 95.00 | **100.00** | 95.00 | **100.00** | **100.00** | 97.50 | **100.00** |
|  | 16 | 96.00 | **100.00** | 99.00 | 78.00 | **100.00** | 80.00 | 97.50 | **98.33** | 87.50 | 87.71 |
|  | 32 | 98.00 | **99.00** | 98.00 | 80.00 | **99.00** | 53.75 | 81.98 | **91.04** | 56.88 | 55.21 |
| Loop | 8 | 12.00 | **100.00** | **100.00** | 98.00 | **100.00** | **100.00** | **100.00** | 97.50 | 82.50 | 85.83 |
|  | 16 | 10.00 | 99.00 | **100.00** | 99.00 | 99.00 | 53.75 | **100.00** | 96.25 | 67.50 | 59.79 |
|  | 32 | 11.00 | 99.00 | **100.00** | **100.00** | 99.00 | 28.12 | **98.85** | 72.50 | 38.75 | 27.29 |
| Sequence | 8 | **100.00** | 99.00 | 78.00 | 98.00 | **100.00** | 95.00 | 99.17 | 99.17 | **100.00** | **100.00** |
|  | 16 | 96.00 | 95.00 | 73.00 | 89.00 | **100.00** | 83.75 | 93.54 | **97.50** | 90.00 | 94.58 |
|  | 32 | **98.00** | 68.00 | 51.00 | 84.00 | 98.00 | 48.12 | 86.15 | **93.75** | 64.38 | 64.27 |
| Signal | 8 | 0.00 | **100.00** | **100.00** | 99.00 | **100.00** | **100.00** | **100.00** | **100.00** | 98.75 | **100.00** |
|  | 16 | 0.00 | **100.00** | **100.00** | 97.00 | **100.00** | 57.50 | **100.00** | 91.25 | 95.00 | 93.54 |
|  | 32 | 1.00 | 90.00 | **97.00** | 64.00 | **97.00** | 17.50 | **83.12** | 68.12 | 75.94 | 48.85 |

Table 9: Finish Rate and Safety Rate of **STLPY** [17]: (multi-agent), **STLPY(S)** [17] : STLPY with a single common plan, **CE** [46] : Centralized Counterexample guided planner and our approaches (**ODE**, **GNN-ODE**). vs the number of agents ($N$) and specification complexity for the DubinsCar Environment. Note that a successful trajectory is both finishing the task *and* being safe. While STLPY and other single agent planning schemes may reach the goal quickly, safety is violated (shown in the Safety Rate). The results are averaged over 30 seeds and we highlighted the best result in **bold**.

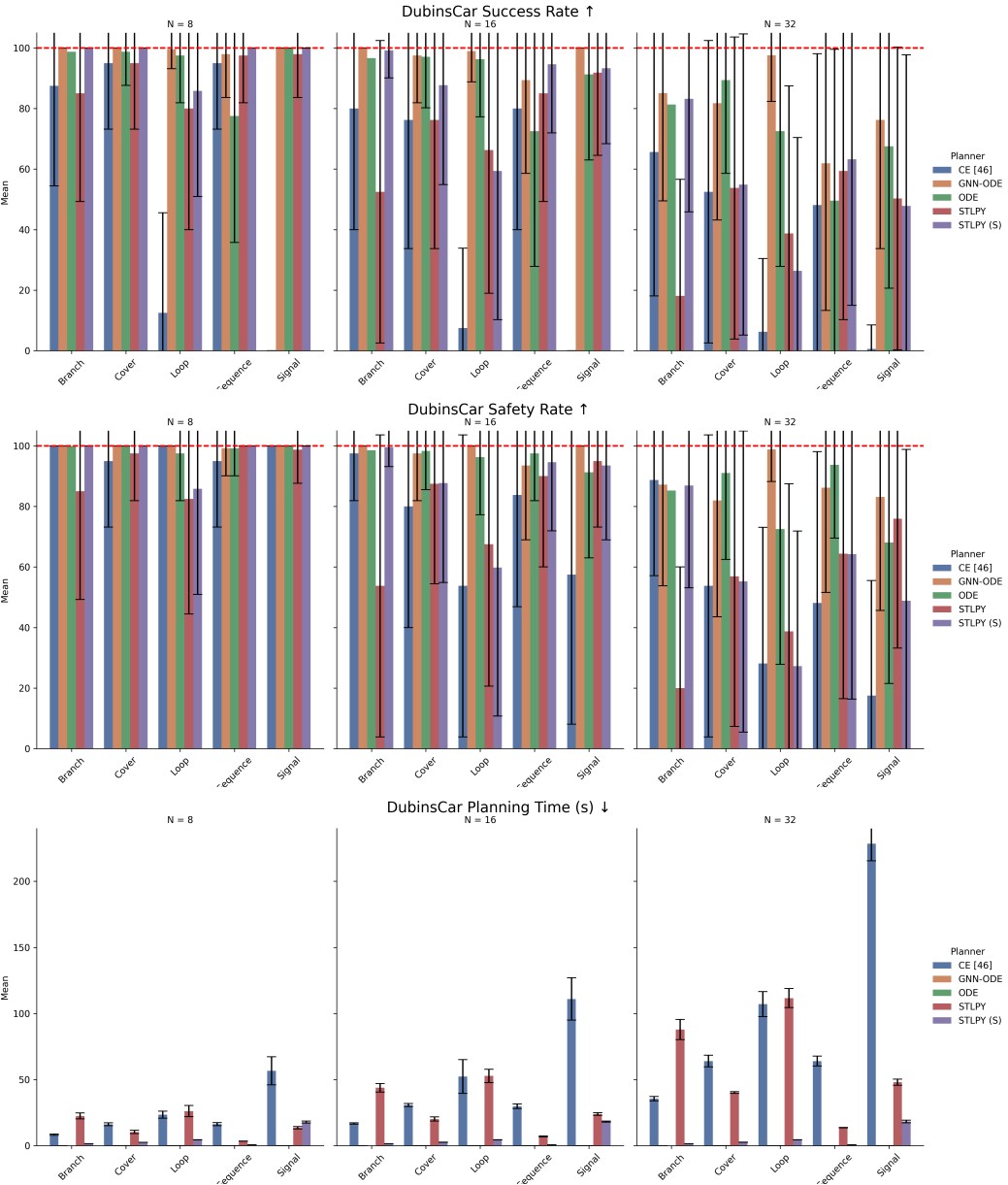

Figure 2: We provide bar plots with error bars noting the standard deviations of the metrics considered. These are complementary to Table 8 and Table 9.

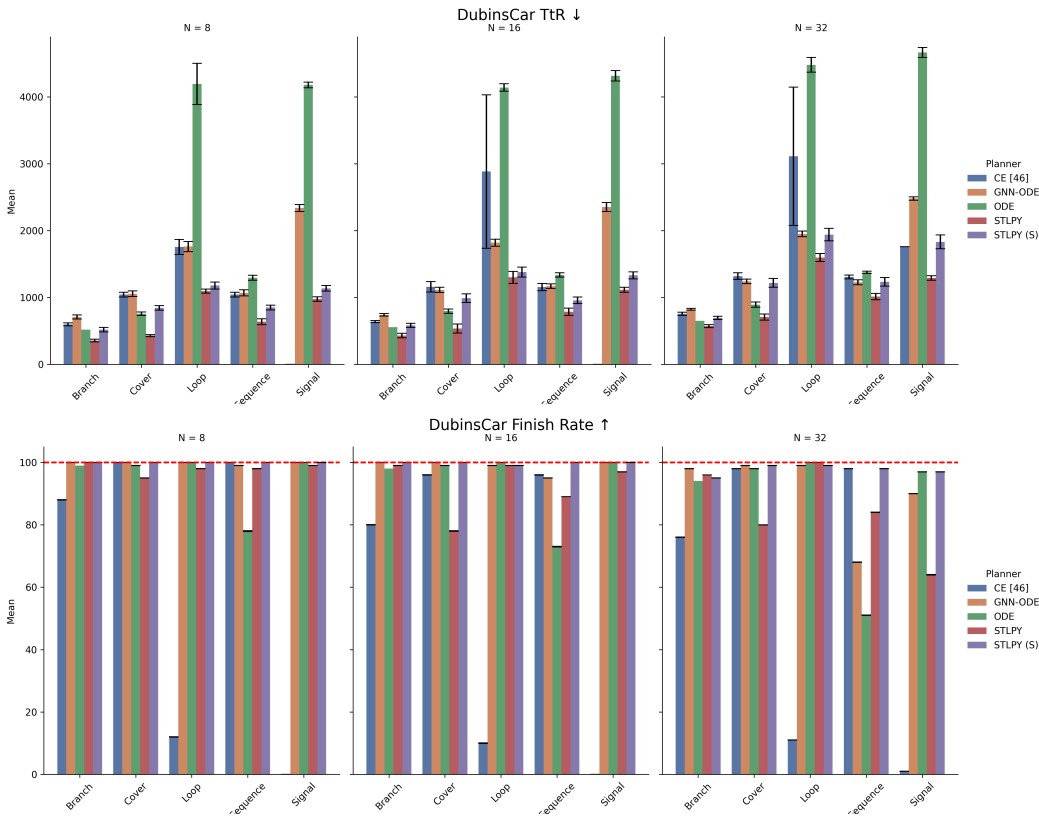

Figure 3: We provide bar plots with error bars notating the standard deviations of the metrics considered. These are complementary to Table 8 and Table 9.

