# OpenReview forum: "Scaling Safe Multi-Agent Control for Signal Temporal Logic Specifications"
_robot-learning.org/CoRL/2024/Conference — CoRL 2024_

### Official Review · Reviewer_uqbs · 2024-07-20

**Originality:** 2
**Technical Quality:** 4
**Clarity Of Presentation:** 3
**Potential Impact:** 3
**Recommendation:** 3
**Confidence:** 4

**Review:**

The paper reads well overall, but there are some opportunities for confusion since the proposed framework has multiple parts that are discussed individually before being brought together in Section 4.4. Consider adding more high-level introduction to your framework earlier on to avoid the risk of losing your reader.

This paper does a good job of combining different building blocks, some known (GCBF, differentiable STL) and some new (GNN+ODE planner), into a useful system.

Overall, I think this paper would be well-received at CoRL, but there are a number of minor things the authors should be sure to address. These include:

Section 1.1: there is an additional class of STL planners that attempt to improve scalability over MILP planners by using nonlinear optimization (sacrificing completeness but improving efficiency). See e.g. [C. Dawson, C. Fan/ "Robust Counterexample-guided Optimization for Planning from Differentiable Temporal Logic." IROS 2022](https://arxiv.org/pdf/2203.02038) . You also cite [19] and [40] later in the paper; perhaps those should be mentioned in Section 1.1 as well?

The problem setting assumes that the STL requirements for each agent are independent; is this a restrictive assumption? This would seem to preclude potentially useful specifications like "Agent 1 remains in area A until agent 2 gets there".

Fig. 1 is a bit dense; it's a lot of information before the reader really catches up on what you're doing. Have you considered splitting it up so that each subfigure is close to the relevant section of the paper?

Can you provide error bars for Table 2?

Can you report the total training time in addition to the planning time?

You claim that your approach is decentralized, but doesn't evaluating the GNN require communication between nodes (message passing)?

I can see an additional limitation in this approach not mentioned in the limitations section: relative to single-shot planners (e.g. using differentiable MA-STL to directly optimize the goals), your approach will require additional samples and time to train the GNN. This will be earned back if you have to plan multiple times in the same environment and with the same specification, but is lost if the specification changes.

I notice that your approach has a high optimality gap (in time to reach) relative to MILP planner. Is that because the MILP sacrifices STL satisfaction for time to reach, or is it a limitation of your method?

A number of details about the training process seem to be missing (at least, I couldn't find them). Such as number of randomly sampled initial conditions per epoch, number of epochs, hyperparameters, .... These should be included in the supplementary material so that others can reproduce your work.

You also conduct hardware experiments, but hide them in the supplementary materials. The CoRL audience generally values hardware demonstrations, so make sure to highlight them!

**Quality Of The Limitations Section:**

2

**Questions For Rebuttal:**

Please respond to my questions in the review above.

**Robotics Focus:**

4

**Summary Of Paper:**

This paper proposes a hierarchical architecture for multi-agent control with STL constraints, wherein a GNN and recurrent MLP predict a set of high-level waypoints that are tracked using a neural graph control barrier function controller. This combination is trained end-to-end on different environments and achieves better empirical scalability than MILP-based solution methods.

**Summary Of Recommendation:**

I think this paper would be well-received at CoRL, but there are a number of minor things the authors should be sure to address.

---

### Official Review · Reviewer_rCpB · 2024-07-23
**Clear writing but limited applicability of the proposed method.**

**Originality:** 3
**Technical Quality:** 3
**Clarity Of Presentation:** 3
**Potential Impact:** 2
**Recommendation:** 3
**Confidence:** 3

**Review:**

Overall, I think the writing is clear, and within the scope of the problem that the authors specified, the method seems to work well. However, since the method requires a rather strong assumption that the agents share common dynamics and STL objectives, I think the applicability of the work is quite limited. Other than this, I only have minor comments listed below:
- In definition 1, Achievability - the tracking performance requirement is written in sum-over-time form. Is there a particular reason for this or could this be written in other forms? (for instance, tracking error not exceeding the threshold for all time instances)
- In Section 4.2, since the sequence of goals is produced by a forward pass of the MLP neural network, it seems like the error will compound through these integration steps.
- Results in Table 2: in line 271, it is said that 5 random initial seeds are sampled for this statistic. Is it reliable enough?
- The planner and the GCBF controller are learned together. Is there any specification of the update rates between the two entities? For instance, are they updated in the same frequency and same learning rates?

**Quality Of The Limitations Section:**

3

**Questions For Rebuttal:**

The author claims that by decoupling the problem of STL specification and safety, the proposed method does not suffer the poor scalability that other methods that solve STL suffer. However, in order to achieve this, the method operates under a strong assumption of homogeneous dynamics and objectives. Under this strong assumption, wouldn’t other methods’ scalability also improve?

**Robotics Focus:**

4

**Summary Of Paper:**

This paper concerns a problem of multi-agent planning where each agent shares the same dynamics and the STL specifications of their objectives. The problem is decoupled into two learning problems, one that learns the neural network planner that learns to achieve the STL objective, and the other that learns the Graph CBF to ensure that each agent does not collide to each other and the obstacles.

**Summary Of Recommendation:**

The method only works under strong assumptions, however, under this setting, I think the presentation of the paper and its originality are both very clear.

---

### Official Review · Reviewer_HZ5o · 2024-07-28
**Multi-agent STL planner**

**Originality:** 2
**Technical Quality:** 3
**Clarity Of Presentation:** 4
**Potential Impact:** 3
**Recommendation:** 3
**Confidence:** 5

**Review:**

The paper is well-motivated, as mixed-integer-based planners often suffer from the curse of dimensionality, especially in multi-agent settings. Moreover, signal temporal logic (STL) can capture much richer mission specifications compared to traditional planning methods. However, the paper assumes that all agents receive the same type of temporal logic formula throughout the entire mission. In my opinion, this is quite a limitation, as we often want agents to perform different tasks. If all agents have the same STL formula, one could plan for a single agent and then use low-level abstraction controls to steer the group of agents (while maintaining formation) to achieve the same objective. The experimental results show promising speed improvements across all scenarios, with the maximum number of agents tested being 32. The paper is well written, but some questions need to be addressed for further clarity and improvement. The hardware experiment is appreciated but the sequence is not that exciting, a more convincing scenario is needed for the drone demonstration.

**Quality Of The Limitations Section:**

2

**Questions For Rebuttal:**

(1) Why is safety treated separately from STL satisfaction? One can directly specify safety and liveness through an STL formula, which can be encoded as part of robustness, i.e., a measurement of how safe the agents are.

(2) The "co-learning" of the safety (GCBF+) and objective controllers (GNN-ODE) is not clear to me. Are there cases where the GNN-ODE controller and GCBF+ controller cause the planner to be unable to find a solution? If yes, how do you address this?

(3) In Table 2, the STLPY's safety rate is decreasing dramatically. Can you explain this in more detail? If your method is using GCBF+ and GNN-ODE, is the baseline algorithm also using some sort of CBF-based safety checking mechanism?

(4) The STL formula used for scenarios (seq, cover, loop, branch) does not contain the "Until" operator. Can you show an example that includes it?

**Robotics Focus:**

4

**Summary Of Paper:**

The paper proposes a multi-agent planning framework using Signal Temporal Logic. The proposed method relies on a graph neural network to generate plans in a decentralized manner. A comparison against mixed-integer linear programming-based planning is made.

**Summary Of Recommendation:**

The motivation of the paper is good and I appreciate the hardware experiments. Some technical details need to be addressed as it directly impacts the claim from the authors.

---

### Author Rebuttal · Authors · 2024-08-14

We are grateful to the reviewers for their contributions to refining our work. Below we present experimental results to address concerns by the reviewers. We focus on the most complex non-linear environment as presented in the main paper (DubinsCar).

---

### Decision · Program_Chairs · 2024-09-04

**Decision:**

Accept

**Comment:**

This paper received 3 confident reviews. It proposes a hierarchical architecture for multi-agent control with STL constraints, wherein a GNN and recurrent MLP predict a set of high-level waypoints that are tracked using a neural graph control barrier function controller. This combination is trained end-to-end on different environments and achieves better empirical scalability than MILP-based solution methods.

To address in rebuttal (in addition to reviewer comments):
- Discuss limitations on homogeneity, noted by several reviewers (dynamics, as well as STL specs)
- Address technical queries highlighted by reviewers
- Add training details
- Discuss insights obtained on real robot experiments

Post-rebuttal:
The reviewers provided key clarifications. The experimental analyses should ideally be extended in support of claims.